# Defining complicated urinary tract infection and route of antibiotics in children presenting to the emergency department: a cohort study using the Melbourne RUPERT clinical score

Barry T Scanlan ,[1,2,3] Laila F Ibrahim,[1,2,3,4] Franz E Babl ,[3,5,6] Sandy M Hopper,[3,5,6] Sarah McNab,[2,3,4] Susan M Donath,[7] Andrew Davidson,[3,8,9] Penelope A Bryant [1,2,3,10]

For numbered affiliations see end of article.

**Correspondence to**
Dr Penelope A Bryant;
penelope.bryant@rch.org.au

## ABSTRACT

**Objectives** Most children with uncomplicated urinary tract infections (UTI) can be managed with oral antibiotics. However, identifying those likely to fail oral and need intravenous antibiotics due to complicating features at presentation is challenging. We aimed to derive, validate and test a score to guide initial antibiotic route.

**Design** This cohort study enrolled children both prospectively and retrospectively. Patients were divided into two groups based on whether they received intravenous or oral antibiotics after 24 hours, including those who switched between routes. Children diagnosed with confirmed UTI were used to derive then validate the score, comparing complicating clinical features between the two groups. Combinations of significantly differentiating features generated receiver operating characteristic curves and the optimal cut-off for intravenous antibiotic use was selected.

**Setting** The emergency department of a tertiary paediatric hospital.

**Participants** All children aged 3 months–17 years with suspected UTI were eligible, and were included if they fulfilled the diagnostic criteria for UTI.

**Outcome measures** The effectiveness of the derived clinical score to differentiate patients at presentation who had complicated UTI requiring ongoing intravenous antibiotics.

**Results** There were 1240 patients, of whom 167 children aged 12 months–11 years with confirmed UTI comprised the derivation cohort. The combination of features that performed optimally (area under curve 0.85, 95% CI 0.79 to 0.91) were: rigors, urological abnormality, pyrexia (≥38°C), emesis, recurrent (≥3) UTI, tachycardia: the RUPERT score (1 point each, maximum 6). A score ≥3 accurately classified route of antibiotics after 24 hours for 80% patients (sensitivity 77%, specificity 81%). For the 168 patients in the validation cohort, the score accurately classified 76% (sensitivity 67%, specificity 78%). The score tested well in 'probable' UTI and adolescents, and less well in infants.

**Conclusion** The Melbourne RUPERT score provides the first standardised, easy-to-use score to aid clinicians in deciding route of antibiotics for more complicated UTI in children. It now needs prospective validation.

---

## STRENGTHS AND LIMITATIONS OF THIS STUDY

⇒ The study design builds on the existing evidence and creates a standardised approach where evidence is lacking.

⇒ The study deliberately incorporates a broad range of children with urinary tract infections to offer straightforward, clinical guidance for which patients with complicating features may benefit from initial intravenous antibiotic treatment.

⇒ The methods ensured internal validation and testing on different patient cohorts.

⇒ The study has the potential limitation that in the absence of an accepted gold standard to determine route of treatment, we have accepted that the treating clinicians made the correct judgement after 24 hours.

⇒ The single-centre study design may not be reflective of all populations or universal practice and the score therefore needs prospective multicentre validation.

---

## INTRODUCTION

Urinary tract infections (UTI) are common in children, with a prevalence rate in by the age 7 years of 1%–2% in boys and 7%–8% in girls.[1 2] Antibiotics are the mainstay of treatment, and most children are successfully treated with oral antibiotics. The decision between intravenous and oral antibiotics is important: not using intravenous antibiotics when they are needed risks clinical deterioration, while overuse places unnecessary burden on children and the healthcare system. Systematic reviews have concluded that oral antibiotics are sufficient for both lower UTI[3] and uncomplicated upper UTI,[4] including with fever, and these recommendations

have been incorporated into the American Academy of Pediatrics (AAP)[5] and UK National Institute for Health and Care Excellence (NICE) guidelines.[6] For complicated UTI, intravenous antibiotics are often—but not always—used. There is no consensus on the definition of complicated UTI: definitions variably include multiple factors related to history and current presentation that may result in divergence from typical management. It is challenging to identify which children with a variety of complicating features at presentation should receive initial intravenous antibiotics because trials have variably excluded those with urological abnormality,[7–10] previous UTI,[7 8] vomiting,[8 10] dehydration[8] or prior oral antibiotics,[7–9] so recommendations are less robust.

In a previous study at our hospital, 72% children presenting to the emergency department had at least one of these complicating features and 28% received intravenous antibiotics.[11] Although there is no universal definition of complicated UTI, preliminary data showed that with an increased number of complicating features, there was a higher likelihood of initial intravenous antibiotic use.[11] When several factors inform treatment choice, clinical scores can help by combining important features.[12] We hypothesised that combining complicating features of UTI would generate a clinical threshold to determine route of antibiotics, and could provide an approach to defining complicated UTI.

We aimed to derive, validate and test a score that combines complicating clinical features and incorporates existing evidence to assist in determining the initial route of antibiotics in ED in children with UTI.

## METHODS

### Study design

A clinical population of consecutive patients presenting to ED with UTI was used to derive, validate and test a clinical score for defining complicated UTI and initial route of antibiotics.

### Setting

The study was conducted in ED at The Royal Children's Hospital Melbourne, a tertiary paediatric hospital, from May 2016 to March 2018.

### Inclusion/Exclusion criteria

All children aged 3 months–17 years inclusive with likely UTI in ED were eligible, and included consecutively if they fulfilled the NICE diagnostic criteria[6] for symptoms and urinalysis (box 1). Patients were enrolled both prospectively to ensure integrity of the information collected, and also retrospectively to include sufficient patients to validate the findings. Patients were excluded if they were treated in ED via superseding pathways: severe infection (eg, sepsis or meningitis) or immunodeficiency (eg, febrile neutropenia or postrenal transplant).

### Clinical procedure and patient outcomes

After consent, data were collected using a standardised case record form during ED assessment for patients

---

**Box 1    Definitions of groups**

**(1) Definitions for eligibility**

**NICE clinical criteria\* for diagnosis of UTI at presentation to ED**

Symptoms: age <2 years—fever, rigor, vomiting, flank tenderness, poor feeding, irritability, offensive urine, gross haematuria or failure to thrive; age ≥2 years—fever, rigor, vomiting, abdominal pain, flank pain/tenderness, dysfunctional voiding, change in urinary continence, dysuria, gross haematuria, frequency, offensive urine or cloudy urine

AND (collected by clean catch, midstream urine, in/out catheter or suprapubic aspirate). Urinalysis— abnormal urinary dipstick leucocyte esterase >1 or nitrite positive OR white cell count >10×10$^6$/L in uncentrifuged urine OR bacteriuria opinion

**(2) Definitions for derivation/validation/test cohorts**

**Confirmed UTI†**

Patients who met the NICE clinical criteria for diagnosis and who were treated as having UTI
AND
Culture positive urine culture with no more than two species of microorganisms >10$^7$ CFU/L OR positive blood culture and no other recognised cause

**Probable UTI**

Patients who met the NICE clinical criteria for diagnosis and who were treated as having UTI but NOT fulfilling research criteria recommendations for culture-based 'confirmed' UTI.

**(3) Definitions for group classification**

'*Likely to fail oral antibiotics*'—complicating features (vomiting, dehydration, urological abnormality‡, previous UTI or oral antibiotics prior to presentation) and on intravenous after 24 hours
'*Oral antibiotics applicable*'—no complicating features, or complicating features and on oral after 24 hours

\*NICE guidelines.[25]
†Trial design recommendations.[25]
‡Urological abnormalities included any known anatomical or functional abnormality of the urological tract, as deemed relevant by the physician in the management of the current UTI. These ranged from moderate/severe (eg, ureteric obstruction, duplex kidneys with VUR, grades III–V VUR, grades III–IV hydronephrosis, neurogenic bladder, postpyeloplasty and postureteric implantation), to mild (eg, posthypospadias repair, grade I–II VUR, grade I–II hydronephrosis, phimosis with balanitis).
CFU, colony-forming units; ED, emergency department; NICE, National Institute for Health and Care Excellence; UTI, urinary tract infection; VUR, vesico-ureteric reflux.

---

enrolled prospectively, and included demographics, clinical features prior to and in ED (table 1), history including urological abnormality (defined in box 1) and urine results. Clinical observations recorded were fever ≥38°C, tachycardia (heart rate >95th centile by age),[13] tachycardia when afebrile and hypotension (systolic blood pressure <5th centile by 50th height centile).[14] Prospective data collected were compared with the electronic medical record and on determining a high degree of concordance, the same data were collected from retrospective participants using the same criteria. The concordance enabled missing clinical features in the retrospective clinical record review to be presumed

**Table 1** Comparison between the 'oral antibiotics applicable' and 'likely to fail oral antibiotics' groups in the derivation cohort

| | Oral antibiotics applicable No. (%) N=132 | Likely to fail oral antibiotics No. (%) N=35 | OR (95% CI) | P value |
|---|---|---|---|---|
| Age, years (mean±SD) | 5.1±3.1 | 4.9±3.2 | | 0.98 |
| Female | 103 (78) | 26 (74) | 0.8 (0.3 to 1.9) | 0.64 |
| Clinical features (either recorded in ED or reported in the previous 24 hours) | | | | |
| Fever ≥38°C | 81 (61) | 32 (91) | 6.7 (2.1 to 22) | <0.001 |
| Rigors | 8 (6) | 6 (17) | 3.2 (1.1 to 10) | 0.04 |
| Vomiting | 41 (31) | 18 (51) | 2.4 (1.1 to 5) | 0.03 |
| Lethargy | 19 (14) | 13 (37) | 3.5 (1.5 to 8) | 0.002 |
| Abdominal pain | 61 (46) | 17 (49) | 1.1 (0.5 to 2.3) | 0.8 |
| Flank pain/tenderness | 16 (12) | 9 (26) | 2.5 (1 to 6.2) | 0.045 |
| Dysfunctional voiding | 5 (4) | 1 (3) | 0.8 (0 to 5.1) | 0.79 |
| Dysuria | 62 (47) | 8 (23) | 0.3 (0.1 to 0.8) | 0.01 |
| Offensive urine | 17 (13) | 3 (9) | 0.6 (0.2 to 2.1) | 0.49 |
| Gross haematuria | 14 (11) | 1 (3) | 0.2 (0 to 1.5) | 0.15 |
| Frequency | 23 (17) | 2 (6) | 0.3 (0 to 1.2) | 0.08 |
| Suprapubic tenderness | 22 (17) | 3 (9) | 0.5 (0.1 to 1.6) | 0.23 |
| Previous history | | | | |
| Urological abnormality* | 20 (15) | 17 (49) | 5.3 (2.4 to 11.9) | <0.001 |
| Any previous documented UTI | 52 (39) | 24 (69) | 3.4 (1.5 to 7.3) | 0.002 |
| Recurrent (≥3) documented UTI | 26 (20) | 21 (60) | 6.1 (2.8 to 13.5) | <0.001 |
| Antibiotic prophylaxis (current) | 6 (5) | 4 (11) | 2.7 (0.8 to 9.6) | 0.13 |
| Known resistant organisms† | 19 (14) | 10 (29) | 2.4 (1 to 5.7) | 0.05 |
| Prior oral antibiotics for UTI | 26 (20) | 8 (23) | 1.2 (0.5 to 2.9) | 0.68 |
| Clinical observations | | | | |
| Tachycardia (any) | 27 (20) | 18 (51) | 4.1 (1.9 to 9) | <0.001 |
| Tachycardia when afebrile | 4 (3) | 4 (3) | 4.1 (1.1 to 16) | 0.04 |
| Hypotension | 0 (0) | 0 (0) | NA | NA |

*Urological abnormality: known anatomical or functional abnormality of the urological tract; this included any abnormality that the treating physician felt was relevant to the UTI management.
†Resistant to the recommended antibiotics in the institutional guideline.
.ED, emergency department; NA, not available; UTI, urinary tract infection.

absent. The decision to start intravenous or oral antibiotics was made by an experienced ED physician (at least registrar or fellow level). Institutional empiric guidelines for intravenous antibiotics for UTI recommend gentamicin administered every 24 hours and benzylpenicillin administered every 6 hours. Ongoing antibiotic route was re-assessed at 24 hours for patients in hospital. Patients discharged from ED on oral antibiotics were presumed to remain on oral unless they represented to our hospital and were switched to intravenous antibiotics. To test the assumption they were not representing to other hospitals, we contacted all prospective patients within 7 days, and none represented to other hospitals. Data were entered into Research Electronic Data Capture[15] at Murdoch Children's Research Institute.

### Standard for assessing treatment

There is no gold standard to determine which children with UTI need intravenous antibiotics. A standard for assessing appropriate antibiotic route was defined using existing evidence and published methodology. For a UTI clinical score to be useful, lower and upper UTI should be included as the distinction can be challenging in young children. Existing studies show children with lower UTI and upper uncomplicated UTI do not need intravenous antibiotics.[3 4] For patients excluded from those studies—those with complications of vomiting, dehydration, urological abnormality, previous UTI or oral antibiotics prior to presentation—a published method for establishing a standard for appropriate route was applied: 'the route of ongoing antibiotic treatment after review by 24 hours is likely to be appropriate based on ongoing symptom progression, even if antibiotics were administered via the other route at presentation'.[16] Therefore, if a patient on intravenous antibiotics was reviewed within 24 hours, and switched to oral antibiotics, they were determined to have only needed oral antibiotics initially (box 1). Likewise, if a patient treated with oral antibiotics represented within 24 hours and was started

on intravenous, they were determined to have needed intravenous antibiotics initially. Based on this standard, patients were allocated to one of two groups: 'oral antibiotics applicable' (no complicating features[4] or complicating features and on oral after 24 hours) or 'likely to fail oral antibiotics' (complicating features and on intravenous antibiotics after 24 hours).

### Study cohorts
The study cohorts included derivation, validation and test cohorts.

### Derivation cohort
To determine which complicating features are important for a clinical prediction score, patients were included aged 12 months–11 years with confirmed UTI—NICE diagnostic criteria plus positive urine or blood culture[17] (box 1). This age range was defined because patients with UTI under the age of 12 months are frequently managed differently,[18] our previous study showed large variation under 12 months,[19] and studies of childhood UTI characteristically have an upper limit of 12 years.[19]

### Validation cohort
To assess whether the score works in a separate group defined in the same way, the 12 months–11 years age group with confirmed UTI was divided in half into the derivation and validation cohorts by assigning patients alternately by date of presentation. Alternate assignation avoided an external change during the study having a biased effect on one group.

### Test cohort
The score was tested on additional patient groups: (1) age 12 months–11 years with probable (not confirmed) UTI,[17] (2) age 3 months–11 months and (3) age 12 years–17 years.

### Statistical analysis
Univariate analysis was used to compare clinical features that were different between the 'oral antibiotics applicable' and 'likely to fail oral antibiotics' groups. For categorical data, Fisher's exact test was used and for continuous data

Student's t-test was applied. ORs and 95% CIs were calculated and $p < 0.05$ was considered statistically significant. Each differentiating feature between the two groups was converted to a binary score of 0 (absent) or 1 (present). Different combinations of these features were added and used to generate receiver operating characteristic (ROC) curves. Scores for each feature were weighted (increased to 2) to determine whether that improved performance. The score performance with different cut-offs for identifying those likely to fail oral antibiotics was assessed as follows: (1) high ROC area under the curve (AUC); (2) most accurate assignment of patients to the groups 'oral antibiotics applicable' and 'likely to fail oral antibiotics' and (3) sensitivity and specificity. The combination of features that had the highest overall results in all of these domains was used to derive a clinical score. Statistical analysis used Stata/IC V.15.1.

### Patient and public involvement
Patients and/or the public were not involved in the design, or conduct, or reporting, or dissemination plans of this research.

## RESULTS
Over 22 months, 1438 patients had a diagnosis of UTI, of which 46 were excluded due to miscoding or other reasons (figure 1A). 1392 children were treated as having a UTI in ED. Of these, 152 were excluded for not fulfilling the NICE criteria (box 1) or other exclusions, resulting in 1240 patients with UTI. Out of these 1240 cases, 276 (22%) were aged 3 months–11 months, 831 (67%) were aged 12 months–11 years and 133 (11%) were aged 12 years–17 years.

### Clinical features and treatment of derivation cohort
Of 831 patients aged 12 months–11 years, 335 had confirmed UTI (box 1). Half (167) were assigned to the derivation cohort and 168 to the validation cohort, with no difference between groups in triage category and hospital admission (data not shown). The treatment allocations at 24 hours were: derivation cohort 35 (21%) intravenous

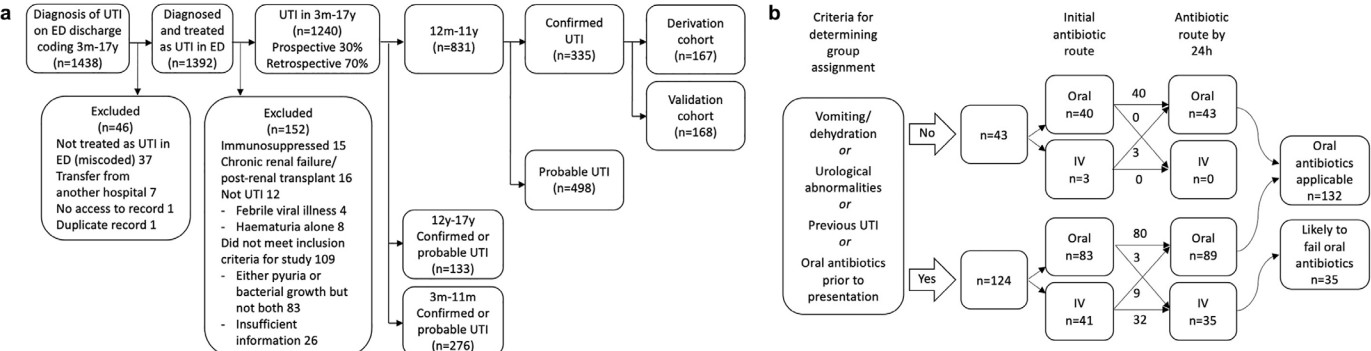

**Figure 1** Patient assignment to (A) derivation, validation and test cohorts and (B) within the derivation cohort to 'oral antibiotics applicable' and 'likely to fail oral antibiotics' groups in the derivation cohort. ED, emergency department; m, month(s); y, year(s); UTI, urinary tract infections.

and 132 (79%) oral; validation cohort 47 (28%) intravenous and 121 (72%) oral. In the derivation cohort, there were 43 children with no complicating features and 89 with complicating features who were on oral by 24 hours (80 continuing and 9 de-escalated from intravenous) totalling 132 in the 'oral antibiotics applicable' group. Of the nine who de-escalated, eight had only a 24-hourly dosed antibiotic and one additionally had multidose antibiotics per day. There were 35 children with complicating features and on intravenous after 24 hours (32 continuing and 3 escalated from oral) and these were assigned to the 'likely to fail oral antibiotics' group (figure 1B).

### Derivation of clinical score

Multiple patient characteristics and clinical features that have been variably included in definitions of complicated UTI or excluded from definitions of uncomplicated UTI were compared between the 'oral antibiotics applicable' and 'likely to fail oral antibiotics' groups. Ten features significantly differentiated between the groups (either recorded in ED or reported in the previous 24 hours): fever ≥38°C, rigors, vomiting, lethargy, flank pain/tenderness, dysuria, urological abnormality (known anatomical or functional anomaly of any part of the urological tract), previous UTI, documented recurrent (≥3) UTI, any tachycardia, tachycardia when afebrile (table 1). For features that were variants of the same measure, the one that differentiated the groups more significantly was selected: 'any tachycardia' differentiated more than 'tachycardia when afebrile' and 'recurrent (≥3) UTI' differentiated more than 'any previous UTI'. The ROC curve of this eight-feature score had a high AUC of 0.86 (95% CI 0.81 to 0.92). Features were serially removed from this score and different cut-offs were tested to compare performance in terms of AUC, correct identification of 'likely to fail oral antibiotics', sensitivity and specificity. Score performance did not deteriorate until the number of features was below six (online supplemental table 1 and figure 2A). There was expected overlap between some features: for example, in the total population, 217 (18%) children had both fever and tachycardia, while 92 (7%) had tachycardia when afebrile and 444 (36%) had fever

without tachycardia. Likewise, 130 (6%) had both urological abnormality and recurrent UTI, while 214 (10%) patients had urological abnormality without recurrent UTI and 261 (12%) patients had recurrent UTI without documented abnormality.

### The Melbourne RUPERT score

Using six of the eight discriminating features, the combination that performed best was rigors, urological abnormality, pyrexia, emesis, recurrent (>2) UTI and tachycardia, which can be remembered using the acronym RUPERT. Applying different weights did not affect score performance (data not shown). The score had a ROC AUC of 0.85 (95% CI 0.79 to 0.91) (figure 2A). Next, a RUPERT score cut-off was determined, above which patients should be considered for intravenous antibiotics due to having more complicated UTI, and below which oral antibiotics could be used. A cut-off of 3—patients with a score of 3 or more considered for intravenous antibiotics—correctly classified the route of antibiotics after 24 hours for 80% of patients, with sensitivity 77% and specificity 81% (online supplemental table 1, figure 3). Applying a cut-off of 3, intravenous antibiotics would have been recommended for 52 (31%) patients and oral for 115 (69%) patients. This threshold for use of intravenous antibiotics may act as a surrogate for identifying more complicated UTI. A cut-off of 4 correctly classified 81% of patients, with sensitivity 31% and specificity 95%. With a cut-off of 4, intravenous antibiotics would have been recommended for 18 (11%) patients and oral for 149 (89%) patients. Increasing the cut-off from 3 to 4 would result in increased patients failing oral treatment from 8 (5%) to 24 (14%). For the nine who de-escalated from intravenous to oral antibiotics by 24 hours, two had a score of 1 (both recurrent UTI), four scored 2 (two fever and vomiting, one fever and tachycardia, one fever and urological abnormality), one scored 3 (fever, rigors and vomiting) and two scored 4 (both fever, vomiting, recurrent UTI and urological abnormality—one with neurogenic bladder, one with pyeloplasty).

To ensure the derived score performed rationally, we examined its use on afebrile patients since only a minority

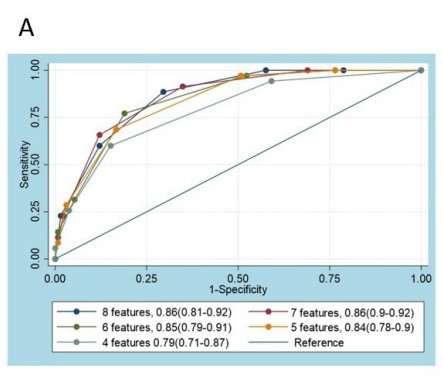
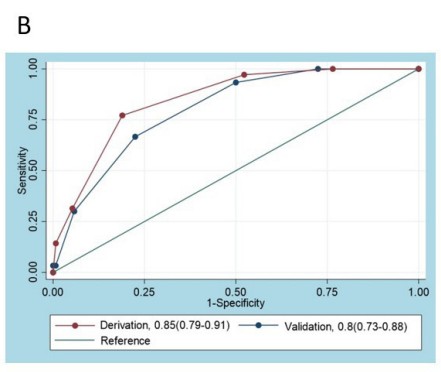
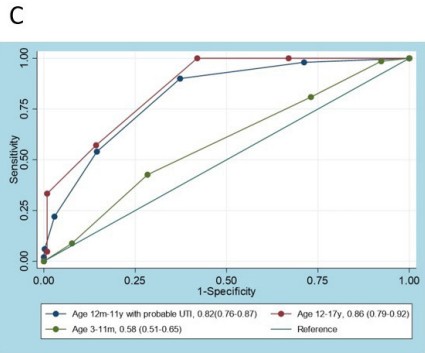

**Figure 2** Comparison of receiver operating characteristic curves with (A) different numbers of features for the derivation cohort; (B) for derivation and validation cohorts and (C) for further test cohorts (area under the curve (95% CI)).

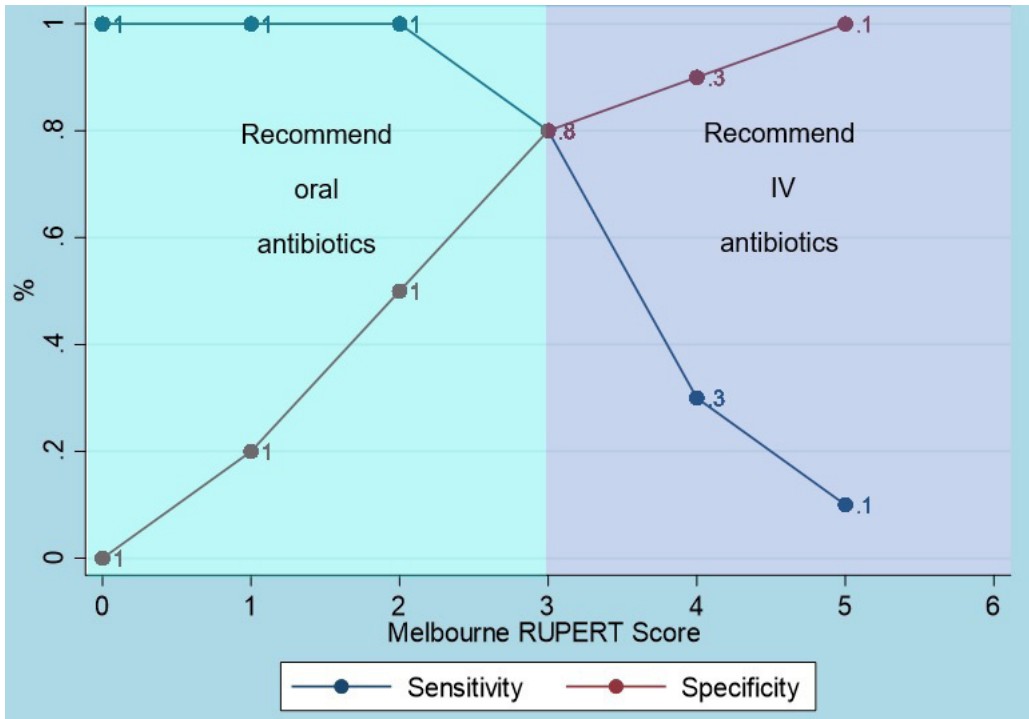

**Figure 3** Sensitivity and specificity at each cut-off threshold of the Melbourne RUPERT score for the derivation cohort. IV, intravenous.

would be expected to need intravenous antibiotics (and these patients would need three other features to reach the threshold). Only three patients were still on intravenous antibiotics at 24 hours: two with a RUPERT score of 3 (both with severe urological abnormalities—neurogenic bladder, Mitrofanoff and severe vesico-ureteric reflux—and recurrent UTI, one also had vomiting and the other also had rigors). The other patient had a RUPERT score of 1 (recurrent UTI), and had 7 days of failed oral treatment, so was treated with intravenous antibiotics while waiting for culture results at 48 hours, at which point was switched to a different oral choice for a resistant pathogen. It was therefore deemed to work appropriately.

## Validation of the RUPERT score

The score was validated on 168 patients with the same criteria (age 12 months–11 years with confirmed UTI) (online supplemental table 2). Applying the same standard, by 24 hours 121 (72%) were receiving oral antibiotics and 47 (28%) intravenous antibiotics. The RUPERT score had a similarly high AUC of 0.80 (95% CI 0.73 to 0.88) (figure 2B) in this cohort. A score cut-off of 3 correctly classified 76% of patients (sensitivity 67%, specificity 78%). Applying a cut-off of 3, intravenous antibiotics would have been recommended for 51 (30%) patients, while oral antibiotics would have been recommended for 117 (70%) patients. With a cut-off of 3 in the validation cohort, 10 (6%) patients who were recommended for oral antibiotics would experience treatment failure (table 2).

### Testing the RUPERT score on other cohorts

1. *Age 12 months–11 years with probable UTI:* there were 498 patients in this group diagnosed with UTI, which was likely but not confirmed by culture.[17] Applying the score resulted in an AUC of 0.82 (95% CI 0.76 to 0.87), and the cut-off of 3 correctly assigned 82% of patients (sensitivity 54%, specificity 85%) (table 2) (figure 2C and online supplemental table 3).
2. *Age 12 years–17 years with confirmed or probable UTI:* there were 133 patients in this adolescent group. Applying the score resulted in an AUC of 0.86 (95% CI 0.79 to 0.92), and the cut-off of 3 correctly assigned 86% of patients (sensitivity 57%, specificity 86%).
3. *Age 3 months–11 months with confirmed or probable UTI:* there were 276 patients in this infant group. Applying the score resulted in an AUC of only 0.58 (95% CI 0.51 to 0.65), and the cut-off of 3 correctly assigned 64% patients—the lowest proportion in any group (sensitivity 43%, specificity 72%).

There was no difference between the performance of the score in the 12 months–17 years age range between the 276 patients where data were collected prospectively (AUC 0.84, 95% CI 0.79 to 0.87) and the 688 patients where collection was retrospective (AUC 0.83, 95% CI 0.79 to 0.87).

## DISCUSSION

The Melbourne RUPERT score is applicable to a broad range of children presenting to hospital with UTI. By including all children presenting to ED with UTI, it avoids

**Table 2** Impact on patient management across all cohorts aged 3 months–17 years

| Cohort | Derivation 12 months–11 years Confirmed UTI No. (%) | Validation 12 months–11 years Confirmed UTI No. (%) | Test 12 months–11 years Probable UTI No. (%) | Test 12 years–17 years All UTI No. (%) | Test 3 months–11 months All UTI No. (%) | Final 12 months–17 years All UTI No. (%) |
|---|---|---|---|---|---|---|
| Patients | 167 | 168 | 498 | 133 | 276 | 966 |
| Route of initial antibiotic treatment | | | | | | |
| Oral | 124 (74) | 117 (70) | 421 (85) | 91 (68) | 153 (55) | 753 (78) |
| Intravenous | 43 (26) | 51 (30) | 77 (15) | 42 (32) | 123 (45) | 213 (22) |
| Route of antibiotic treatment by 24 hours | | | | | | |
| Oral | 132 (79) | 138 (82) | 448 (90) | 112 (84) | 197 (71) | 830 (86) |
| Intravenous | 35 (21) | 30 (18) | 50 (10) | 21 (16) | 79 (29) | 136 (14) |
| Melbourne RUPERT score assignation | | | | | | |
| Recommend oral (score <3) | 115 (69) | 117 (70) | 406 (82) | 105 (79) | 188 (68) | 743 (77) |
| Recommend intravenous (score ≥3) | 52 (31) | 51 (30) | 92 (18) | 28 (21) | 88 (32) | 223 (23) |

UTI, urinary tract infection.

the need to determine whether a child fits an existing definition of complicated UTI. It offers straightforward, clinically relevant guidance for assessing which patients with complicating features of UTI may benefit from initial intravenous antibiotic treatment that is currently not articulated in major guidelines. It also offers a potential new approach to defining complicated UTI: instead of a one-size-fits-all definition based on specific features, it can be considered as a threshold of multiple complicating features that lead to a distinct management pathway. The definition of complicated UTI is widely variable between studies and guidelines with a range of different host conditions and types of severity falling under the same umbrella term. The RUPERT score solves the problem of separately categorising these often overlapping presentations, by using patients' own various complicating features to establish a threshold for recommending intravenous antibiotics, irrespective of the specific individual features. Each of the features in the score is easily assessable, well-defined and objective and the score does not require patients to verbalise symptoms. Additionally, it does not rely on invasive blood tests or waiting for urine culture results. The score performed well in preschool and primary school age children and adolescents with confirmed and probable UTI. It less accurately identified infants under 12 months receiving oral antibiotics. Although this could be attributable to the score having reduced performance in infants, it is more likely due to the frequent initiation and/or continuation of intravenous antibiotics beyond 24 hours in this age group, often without a clear clinical indication. There is no pathophysiological reason that infants aged 3–11 months should be different to older children, but both our data (45% intravenous initiation for all UTI in 3–11 months vs 28% in the derivation/validation cohorts) and previous studies have shown higher intravenous rates in this age group without clear clinical reasons.[18 20] This therefore would benefit from prospective application and testing, and could help reduce unnecessary intravenous in this age group.

Using a RUPERT score cut-off of 3 provides a higher sensitivity to reduce oral antibiotic failures, at the expense of considering more patients for intravenous antibiotics. Alternatively, a cut-off of 4 substantially increases the number of patients recommended for oral antibiotics but reduces the sensitivity. Like most clinical scores, it performs well at the extremes and the mid-range involves a balance between sensitivity and specificity.[12] Scores in the mid-range give clinicians information about the need for monitoring in the first 24 hours of treatment. Scores in the lower range reassure that even with features potentially associated with sepsis (rigors, tachycardia), in isolation do not necessitate intravenous antibiotic use. We did not seek to exclude features that were more likely to occur together (eg, fever and tachycardia, urological abnormality and recurrent UTI), as co-occurrence may signal increased severity of illness or risk of rapid damage to the kidney, when the intravenous antibiotics would be appropriate.

The lack of a gold standard required the development of one from a combination of existing evidence (which is robust due to multiple RCTs contributing), and where evidence was limited, using published methodology derived from early clinical reviews. Although the methodology was devised for cellulitis, we have previously shown that patients with UTI who received <24 hours' intravenous antibiotics were more similar to patients who received only oral antibiotics than patients who received 2–3 days' intravenous antibiotics.[11] This adds weight to the supposition for the devised gold standard that this group only needed oral antibiotics. However, without a definitive clinical marker for 'need for intravenous antibiotics', this could not be guaranteed in every case. It was therefore important to ensure that clinical practice at our hospital is similar to practice elsewhere. A recent study of

UTI at 36 US hospitals between 2010 and 2016 reported that in children aged 3–24 months, 35% were treated with intravenous antibiotics.[21] This is similar to 39% use in this age group at our hospital (data not shown separately).[11]

Guidelines have proved useful in standardising management of UTI, and major international guidelines developed by NICE[6] and AAP[5] are clear in many areas of diagnosis and management (eg, need for imaging). However, both guidelines leave treatment with intravenous antibiotics more flexible, suggesting that it should be based on 'practical considerations' (AAP) and 'if oral antibiotics cannot be used' (NICE). The range of conditions which broadly and inconsistently fall under the wide definition of complicated UTI means that it has consistently been excluded from trials, guidelines and even retrospective studies of management. Clinicians are left trying to determine if their patient really has a complicated UTI, what exactly that means and how to manage the child in front of them. By defining patients not previously fitting within evidence-based recommendations, the RUPERT score adds clinical context to practical considerations and standardises recommendations for when oral antibiotics are more likely to fail. It allows for an individualised approach by using an individual child's own clinical features rather than trying to impose a particular definition of complicated UTI, which is additionally lacking evidence for management.

Incorporating a clinical score into management decisions offers an aid to reducing variation in care. Paediatric scores that have been successfully developed and implemented include the Westley *et al* score[22] for croup, the Pediatric Asthma Severity Score[23] and the Melbourne ASSET score for cellulitis.[16] In addition to reducing variation in care, a clinical score can standardise route of antibiotics in trials, important for generalising outcomes. For example, in one study describing the outcomes of outpatient parenteral antibiotic treatment for UTI, the authors admitted that the patients could have been treated with oral antibiotics.[18] This clinical score could be used for inclusion/exclusion criteria in studies (eg, whether participants exceed a score cut-off) and/or to describe the relative clinical severity of study subjects by recording their RUPERT score.[24]

The strengths of this score, in addition to being the first to attempt to provide standardised guidance where evidence is lacking, are that it is easy to use and does not require invasive tests or waiting for culture results. In addition, by being applicable to all children with UTI because it incorporates existing strong evidence for those with uncomplicated UTI, clinicians do not need to determine whether a patient has complicated UTI before applying it. There are several limitations, the main one being that it has not yet been tested prospectively in other settings and populations. Although the heterogeneity of our population provides some assurance about generalisability, the score could perform differently in other populations that are more homogenous or heterogenous in a different way. Prospective validation in other settings is

therefore essential. Another potential limitation is that we have accepted that the treating clinicians made the correct judgement at 24 hours on treatment route. While these decisions are made by senior doctors, practice varies between clinicians. Review of medical records, however, did not indicate any unusual practice or a high rate of patients returning after receiving oral antibiotics. Using this score does not negate the possibility of overtreating some patients with intravenous antibiotics and undertreating some with oral antibiotics. However, it aims to be a starting point for standardising the rationale for intravenous antibiotics, while allowing institutional flexibility to determine the treatment outcomes for which they are aiming.

## Conclusion

The Melbourne RUPERT score is an easy-to-use and reproducible clinical score for UTI management in children. It is inclusive of all children with UTI, by incorporating robust evidence for children with uncomplicated UTI, and a new approach to defining patients with complicated UTI to standardise risk stratification and treatment recommendations, both for clinical practice and research. As with any decision-support tool, the aim is to offer a guide rather than completely replace clinical judgement for individual patients. The score will now benefit from external validation, refinement and impact analysis.

**Author affiliations**
[1]Hospital-in-the-Home Department, The Royal Children's Hospital Melbourne, Parkville, Victoria, Australia
[2]Clinical Infections, Murdoch Children's Research Institute, Parkville, Victoria, Australia
[3]Department of Paediatrics, University of Melbourne, Parkville, Victoria, Australia
[4]Department of General Medicine, The Royal Children's Hospital Melbourne, Parkville, Victoria, Australia
[5]Emergency Department, The Royal Children's Hospital Melbourne, Melbourne, Victoria, Australia
[6]Emergency Research, Murdoch Children's Research Institute, Parkville, Victoria, Australia
[7]Clinical Epidemiology and Biostatistics Unit, Murdoch Children's Research Institute, Parkville, Victoria, Australia
[8]Department of Anaesthetics, The Royal Children's Hospital Melbourne, Parkville, Victoria, Australia
[9]Melbourne Clinial Trials Centre, Murdoch Children's Research Institute, Parkville, Victoria, Australia
[10]Infectious Diseases Unit, Department of General Medicine, The Royal Children's Hospital Melbourne, Parkville, Victoria, Australia

**Acknowledgements** We would like to acknowledge the participation of patients and families at RCH and clinical staff in the ED.

**Contributors** BTS was involved in the design and co-ordinated the study, carried out the initial and subsequent data analysis, drafted the initial manuscript, revised subsequent drafts and approved the final manuscript as submitted. PAB, FEB, AD, LI, SMcN and SMH were involved in the concept and design of the study, provided input into data analysis, reviewed and revised the manuscript and approved the final draft. SMD was involved in the design of the study, advised on statistical analysis, revised and approved the final manuscript as submitted. All authors take responsibility for the integrity of the data and the accuracy of the data analysis. All authors approved the final manuscript. PAB is responsible for the overall content as the guarantor.

**Funding** This study was made possible through funding from The Royal Children's Hospital Foundation, with wholehearted support from Jim Carroll and the Donald

Ratcliffe and Phyllis Macleod Trust. PAB was supported in part by a Melbourne Campus Clinician-Scientist Fellowship, Melbourne, Australia and in part by a Medical Research Futures Fund Investigator Grant (MRF1197970). FEB was supported in part by a grant from the RCH Foundation and a Melbourne Campus Clinician-Scientist Fellowship, Melbourne, Australia and a National Health and Medical Research Council (NHMRC) Practitioner Fellowship, Canberra, Australia. The emergency research group, MCRI, is in part supported by an NHMRC Centre for Research Excellence Grant for Paediatric Emergency Medicine, Canberra, Australia and the Victorian government infrastructure support programme.

**Competing interests** None declared.

**Patient and public involvement** Patients and/or the public were not involved in the design, or conduct, or reporting, or dissemination plans of this research.

**Patient consent for publication** Not applicable.

**Ethics approval** This study was approved by RCH Human Research Ethics Committee (HREC36255). Participants gave informed consent to participate in the study before taking part.

**Provenance and peer review** Not commissioned; externally peer reviewed.

**Data availability statement** No data are available. No additional data available.

**ORCID iDs**
Barry T Scanlan http://orcid.org/0000-0003-2318-4642
Franz E Babl http://orcid.org/0000-0002-1107-2187
Penelope A Bryant http://orcid.org/0000-0002-5262-5323

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
