## [Reviewer comments · BMJ Open]

ARTICLE DETAILS

TITLE (PROVISIONAL)	Defining complicated urinary tract infection and route of antibiotics in children attending Emergency: the Melbourne RUPERT clinical score – a cohort study
AUTHORS	Scanlan, Barry; Bryant, Penelope; Babl, Franz; Ibrahim, Laila; Hopper, Sandy; McNab, Sarah; Donath, Susan; Davidson, Andrew

VERSION 1 – REVIEW

REVIEWER	Tamar Lubell New York-Presbyterian Morgan Stanley Children's Hospital, Department of Emergency Medicine
REVIEW RETURNED	15-Dec-2023

GENERAL COMMENTS	The study authors aimed to derive a score made up of clinical criteria generally considered to be associated with a complicated UTI to develop a threshold score to guide the initial antibiotic route. The authors should be commended for tackling an important clinical question for a population frequently excluded from studies. The question is clinically important to general pediatricians, pediatric emergency medicine physicians, and specialists (nephrologists and urologists). The design is appropriate to answer the study question. However, the age groupings need clarification, and the methods currently need more details regarding the definitions of the predictor variables and how they were measured for the cohorts collected prospectively and retrospectively. Without appropriate age groups and well-defined criteria, I am also concerned that the application of this score would result in suboptimal performance if/when studied in other populations. Furthermore, it would be helpful to better understand what comprises "initial antibiotics" given that the number of IV doses (e.g., single vs multi-dose IV antibiotics in the ED) was not described for those de-escalated from IV. This may be important as it helps the reader understand the potential clinical use and implications. Additional questions and comments below: Abstract: Would consider including the mixed prospective and retrospective design Introduction: Does a good job of describing the gap in the literature, and the significance of the question is clear. Methods: 1. The grouping of a wide range of ages that are generally separated, for example, pre-verbal non-toilet trained with toilet trained and adolescent children, would benefit from clarification as to why the authors felt this was appropriate. Would the model performance change if the grouping were 3 months-2 years and 2 years -12 years and 12-17 years instead?2. The clinical criteria need to be more clearly defined. Was there a
---

	standardized questionnaire for families and clinicians for those collected prospectively, or was this abstracted from the medical record? For example: -Urological Abnormality-Please describe the conditions and grades of hydronephrosis included. -Recurrent UTI-by parental report or physician documented? Was there overlap in those with recurrent UTI and urologic abnormality? -Tachycardia- may be expected with pyrexia, which may lead to a score of 2 in any child with fever on presentation. -How were abdominal pain, flank pain/tenderness, and dysuria assessed in a preverbal population? 3. Was there a standardized manual of operations for data abstraction for those predictors collected retrospectively? 4. Who did the abstraction? Were the assessors for the predictors blinded to the outcome? 5. How was missing data handled? Were they presumed to be absent? 6. What was the standard route of urine collection by age? 7. I question the need for a separate analysis of patients with “probable UTI” using a definition that did not include positive culture. Although the colony count threshold may differ by the collection method and likelihood of contamination, if there is no bacterial growth, how is it a probable UTI? Given that the signs and symptoms of UTI overlap with other disease entities and that the specificity of pyuria alone is suboptimal, understanding the score performance in this population may not be useful. Results 1. Of the 9 children de-escalated from IV to PO, how many doses of IV did they receive in the ED? It would be helpful to understand the clinical characteristics of this group to determine if the designation of “oral antibiotics applicable” is appropriate. If this cannot be ascertained, I would add this as a limitation. 2. How many children with recurrent UTIs had urologic abnormalities? Tables and Figures Table 1. Please define urologic abnormality for the reader in the footnote. Flow diagram-what proportion of children were collected prospectively vs. retrospectively? Discussion 1. Page 12, line 26-I may also add that this study excluded infants from the derivation population, which may be another explanation for the poor performance in addition to the noted variability in care.
--	---

REVIEWER	Samuel Uwaezuoke University of Nigeria Teaching Hospital
REVIEW RETURNED	12-Jan-2024

GENERAL COMMENTS	Reviewer’s comments for manuscript submitted to BMJ Open Title of manuscript: Defining complicated urinary tract infection and route of antibiotics in paediatric patients: a clinical score General comment The authors of this manuscript reported the use of a derived, validated and tested clinical score (RUPERT score) to guide the initial route of antibiotics in children with urinary tract infection (UTI) who presented at an Emergency Department. Given the long-term renal complications associated with poorly treated UTI, it is
--

imperative to initiate the appropriate antibiotic therapy early, using the right antibiotic choice and the right route of administration. This fact underscores the importance of this study. However, there are specific concerns I have about this manuscript which need clarifications/corrections from the authors

Specific comments

1. Abstract: Under 'Objectives', the introductory statement is unclear. I think the authors need to clarify which form of UTI that can be managed with oral antibiotics. I suggest they specify that 'most children with uncomplicated UTI can be managed with oral antibiotics.' Again, the next statement is befuddling as it failed to differentiate 'complicated UTI' from 'UTI with complications'. Both terms are not the same. The title of the manuscript rather suggests that the authors may mean the former ('complicated UTI'). Thus, the statement that 'identifying those likely to fail oral and need intravenous (IV) antibiotics due to complications is challenging' may better read 'identifying those likely to fail oral or need intravenous (IV) antibiotics due to complicated UTI is challenging.' Under 'Design', the study design was not clearly stated. If this was a cohort study, what was the intervention and what outcomes were measured? Under 'Results', the authors reported that 1,240 patients were included. It is not correct to commence the statement with Arabic numerical; the number should have been written in words. More importantly, the authors should clarify what is meant by the finding that a RUPERT score of > 3 guided the commencement of IV antibiotics that accurately classified 80% of the patients. Do they mean accurately classifying them as having complicated UTI? The same clarification is needed for the succeeding statement on 168 patients in the validation cohort. Finally, ensure that your keywords are MeSH-optimized.

2. Introduction: From the fourth to sixth sentences, the authors should clarify whether they meant complicated UTI or complications of UTI. While the former is associated with underlying conditions such as urological anomalies, bladder outlet obstruction, immunosuppression etc., the latter comprises sepsis, recurrence of infection, Urosepsis, renal scarring etc.

3. Methods: The authors stated here that the study design was a clinical cohort of consecutive patients presenting at the Emergency Department. It is presumed that a cohort study is prospective in nature. However, the authors reported that 'to include sufficient patients to validate the findings, participants were also identified retrospectively through diagnostic coding of UTI.' Can they please clarify this? Again, why was immunodeficiency listed as one of the exclusion criteria of the study given that it is one of the features that also defines complicated UTI? Under 'clinical procedure and patient follow-up', what do the authors mean by collecting data from retrospective patients since the study was essentially a prospective one? (Line 5).

4. Results: The authors reported that 'Of the 1,240 with UTI, 276 (22%) were aged 3m-11m, 831 (67%) were 12m-11y and 133 (11%) were 12y-17y'. I assume these were prospectively recruited participants. What then is the number of the retrospective participants mentioned previously? Among the latter, was the diagnosis of UTI similarly made with the NICE diagnostic criteria? Figures 1 and 3 need magnification for clarity and understanding of the findings.

5. Discussion and conclusion: Well presented with no ambiguity

6. References: Although much of the cited publications appear relevant, some are very old publications. For instance, reference

	number 1 may well be replaced by a more recent publication. Again, there is no consistency in writing the citations; while some journal names were written in their index medicus titles, others appeared with their full names.
--	--

REVIEWER	Kjell Tullus Great Ormond Street Hospital for Children, Nephrology
REVIEW RETURNED	28-Jan-2024

GENERAL COMMENTS	The authors have tried to develop a score to help with the decision if to treat children with a UTI with iv or oral antibiotics. They include children with all kinds of UTIs and of virtually all ages. I am sorry but I do not believe that this is a workable concept. The range of different conditions within that spectrum is too wide to make it meaningful. It is difficult to see but an exceptional child with an afebrile UTI that would need anything else but oral antibiotics. This group should thus be excluded from the study. Any child with signs of a potential sepsis, like rigor and tachycardia out of proportion to the temperature, should also be evaluated on an individual basis and most likely in nearly all cases be given iv antibiotics. (And intensive care?) Children with urinary tract malformations are a heterogenous group with mild malformations like mild hydronephrosis to severe posterior urethral valves with markedly impaired kidney and bladder function. These children need an individualised approach taking a number of important variables into account.
--

VERSION 1 – AUTHOR RESPONSE

Reviewer 1

The study authors aimed to derive a score made up of clinical criteria generally considered to be associated with a complicated UTI to develop a threshold score to guide the initial antibiotic route. The authors should be commended for tackling an important clinical question for a population frequently excluded from studies. The question is clinically important to general pediatricians, pediatric emergency medicine physicians, and specialists (nephrologists and urologists).

Thank you.

The design is appropriate to answer the study question.

Thank you.

However, the age groupings need clarification, and the methods currently need more details regarding the definitions of the predictor variables and how they were measured for the cohorts collected prospectively and retrospectively. Without appropriate age groups and well-defined criteria, I am also concerned that the application of this score would result in suboptimal performance if/when studied in other populations.

Thank you for these suggestions. Regarding age groupings we have stated in the Methods:

Derivation cohort: To determine which complicating features are important for a clinical prediction score, patients were included aged 12 m–11y with confirmed UTI – NICE diagnostic criteria plus positive urine or blood culture¹⁶ (box 1). This age range was defined because patients with UTI under the age of 12 months are frequently managed differently¹⁷, our previous study showed large variation under 12 months¹⁸, and studies of childhood UTI characteristically have an upper limit of 12 years¹⁹.

Test cohorts: The score was tested on additional patient groups: 1) age 12m-11y with probable (not confirmed) UTI¹⁶, 2) age 3m-11m, and 3) age 12y-17y.

For the response to this reviewer's follow-on question about the potential for using different age ranges, see below.

Regarding definitions of the predictor variables, in the Methods we stated the following and have now directed the reader to table 1 for the full list of clinical features collected:

‘... and included demographics, clinical features (prior to and in ED (table 1)), previous history including urological abnormality (defined in Box 1) and urine results. Clinical observations recorded were fever $\geq 38^{\circ}\text{C}$, tachycardia (heart rate >95 th centile by age¹²), tachycardia when afebrile, and hypotension (systolic blood pressure <5 th centile by 50th height centile¹³). The same data were collected from retrospective participants.’

In the Results we have added the definitions to these relevant clinical features to specifically address the reviewer’s concern that the score might not be applied correctly to patients as follows:

‘Ten features significantly differentiated between the groups (either recorded in ED or reported in the previous 24 hours): fever $\geq 38^{\circ}\text{C}$, rigors, vomiting, lethargy, flank pain/tenderness, dysuria, urological abnormality (known anatomical or functional anomaly of any part of the urological tract), previous UTI, documented recurrent (≥ 3) UTI, tachycardia, tachycardia when afebrile.’ (The last two are defined in the Methods as above).

We have also updated table 1 to reflect this. Most of the other features are recognisable to clinicians but to save space in the manuscript we could put more detailed definitions of all clinical features in a supplementary table if preferred.

Furthermore, it would be helpful to better understand what comprises “initial antibiotics” given that the number of IV doses (e.g., single vs multi-dose IV antibiotics in the ED) was not described for those de-escalated from IV. This may be important as it helps the reader understand the potential clinical use and implications.

We have added to the Methods: ‘Institutional empiric guidelines for IV antibiotics for UTI recommend gentamicin 24 hourly and benzylpenicillin 6 hourly.’

We have re-analysed the 9 patients who de-escalated to oral by 24 hours and added their antibiotic doses to the manuscript as follows: ‘Of the 9 who de-escalated from IV to oral by 24 hours, 8 had only a 24-hourly dosed antibiotic, and 1 additionally had multi-dose antibiotics per day.’

Additional questions and comments below:

Abstract: Would consider including the mixed prospective and retrospective design.

Agreed. This has been added as follows: ‘This study assessed children (3 months–17 years) treated in the Emergency Department (ED) with UTI, enrolled both prospectively and retrospectively.’

Introduction: Does a good job of describing the gap in the literature, and the significance of the question is clear.

Thank you.

Methods:

1. The grouping of a wide range of ages that are generally separated, for example, pre-verbal non-toilet trained with toilet trained and adolescent children, would benefit from clarification as to why the authors felt this was appropriate. Would the model performance change if the grouping were 3 months-2 years and 2 years -12 years and 12-17 years instead?

We thought about this carefully, also considering (and rejecting) whether to separate pre-school and primary school aged children, as is done in many studies. The reason for separating under 1 year olds from the older age group is that our previous work and other studies suggest that children under the age of 1y are treated differently, as stated in the manuscript. The reviewer is suggesting 3 slightly different age ranges (moving the 1-2 year olds into the infant group) which are also reasonable groupings. We have re-analysed the data, and changing the youngest age group to 3m-2y results in a ROC AUC of 0.61 - this is because the $<1\text{y}$ is poor at 0.58 and 1-2y is 0.77. This amalgamated group results in the score looking as though it performs better in under 1s than it actually does, and worse in the 1-2s, thereby losing important detail and erroneously suggesting a lack of utility in these older pre-verbal children.

2. The clinical criteria need to be more clearly defined. Was there a standardized questionnaire for families and clinicians for those collected prospectively, or was this abstracted from the medical record?

Yes there was a standardised questionnaire for all prospectively enrolled patients, incorporating all the symptoms in table 1. As above, we have added detail to these in the text and table, but could add further detail to these in a supplementary table if the editors would prefer, or add the case record form

to supplementary information. These questionnaires were then compared to the medical record, and because there was a very high degree of concordance, we had confidence in the medical record as a source of clinical features for the retrospectively collected data, for which the same fields were collected as in the questionnaire.

This standardised data collection has been clarified in the Methods as follows: 'After consent, data were collected using a standardised case record form during ED assessment for patients enrolled prospectively' and 'Data collected were compared with the electronic medical record and on determining a high degree of concordance, the same data were collected from retrospective participants using the same criteria.'

For example:

-Urological Abnormality-Please describe the conditions and grades of hydronephrosis included.

Thank you for highlighting this. Urological abnormalities included any known anatomical or functional abnormality of the urological tract including any grade of hydronephrosis, as deemed relevant by the physician to the treatment of UTI. The aim was to keep this as simple as possible, and in line with exclusions from previous RCTs which simply state 'urological abnormality'. We could additionally add a supplementary table detailing the all the urological abnormalities if required: these range from moderate/severe (79%) to mild (21%). We have not separated these patients out to determine what other complicating features they had (if any), but of the ones we have classified as moderate/severe 91/187 (49%) were treated with IV, whereas 10/40 (25%) with mild urological abnormality were treated with IV. We could do further analysis of this group if the editors would prefer.

In response to this query, to enable better context, we have added to the Methods as follows: '...data were collected... and included... previous history including urological abnormality (defined in Box 1)'

We have added to Box 1 as follows: 'Urological abnormalities included any known anatomical or functional abnormality of the urological tract, as deemed relevant by the physician in the management of the current UTI. These ranged from moderate/severe (e.g. ureteric obstruction, duplex kidneys with vesico-ureteric reflux (VUR), grades III-V VUR, grades III-IV hydronephrosis, neurogenic bladder, post pyeloplasty and post ureteric implantation), to mild (e.g. post hypospadias repair, grade I-II VUR, grade I-II hydronephrosis, phimosis with balanitis).'

We would be happy to add further analysis if needed.

-Recurrent UTI-by parental report or physician documented? Was there overlap in those with recurrent UTI and urologic abnormality?

Recurrent UTI were physician documented – this has been added to the text as above.

There was expected overlap in those with recurrent UTI and urologic abnormality, but this was not universal. We have further analysed the data and 261 (12%) patients had recurrent UTI alone, 214 (10%) patients had urological abnormality alone and 130 (6%) patients had both. Although there is some expected overlap, they are separately included in the score as there many more patients who only had one, and having both may to increase the risk to the child or their kidneys.

To provide better context for the reader for both this and the next point, we have added to the Results as follows: 'There was expected overlap between some features: for example, in the total population, 217 (18%) children had both fever and tachycardia, while 92 (7%) had tachycardia when afebrile and 444 (36%) had fever without tachycardia. Likewise, 130 (6%) had both urological abnormality and recurrent UTI, while 214 (10%) patients had urological abnormality without recurrent UTI and 261 (12%) patients had recurrent UTI without documented abnormality.'

We have added to the Discussion as follows: 'We did not seek to exclude features that were more likely to occur together (e.g. fever and tachycardia, urological abnormality and recurrent UTI), as co-occurrence may signal increased severity of illness or risk of rapid damage to the kidney, when the IV antibiotics would be appropriate.'

-Tachycardia- may be expected with pyrexia, which may lead to a score of 2 in any child with fever on presentation.

As shown in the Results and table 1, both 'any tachycardia' and 'tachycardia when afebrile' were assessed and both were associated with IV use. Because, perhaps slightly counter-intuitively, 'any tachycardia' differentiated IV from oral use better than 'tachycardia when afebrile', this was the feature

included in the score. This is stated in the manuscript as follows: 'For features that were variants of the same measure, the one that differentiated the groups more significantly was selected: i.e. tachycardia differentiated more than tachycardia when afebrile'. We have re-analysed the data and by far the highest third component to the clinical score in children who also had fever and tachycardia (either any or when afebrile) was vomiting i.e. all components of being acutely clinically unwell. While a proportion of these children were still treated with oral antibiotics, as with other bacterial infections causing fever, tachycardia and vomiting it would be appropriate to consider initial IV antibiotics, per scoring ≥ 3 on the score. See above response for additions to the Results and Discussion.

-How were abdominal pain, flank pain/tenderness, and dysuria assessed in a preverbal population? Per the NICE guidelines for preverbal children, (Box 1 in the manuscript) abdominal pain and dysuria were not usually assessed in preverbal children. Flank tenderness was assessed by the treating physician, and we have removed 'flank pain' in the pre-verbal UTI definition from Box 1 as this was incorrectly included.

3. Was there a standardized manual of operations for data abstraction for those predictors collected retrospectively?

Data was extracted manually from the electronic medical record by the ED research team using the same case record form fields as for prospective data collection, and entered directly into REDCap. The reliability of the data collected retrospectively is reflected by the similarity of performance of the score on those patients collected prospectively and retrospectively; this was done as a post hoc analysis to ensure the data were reliable – this is stated in the Results. We have clarified in the manuscript as above regarding having a standardised case record form. We could add the case record form as supplementary information if the editors think useful.

4. Who did the abstraction? Were the assessors for the predictors blinded to the outcome?

The data abstraction was done by the first author and ED research nurse team. They also collected data on the initial and 24 hour route of antibiotics, but had no a priori knowledge about which features differentiated IV from oral or would be ultimately end up being used in the score.

5. How was missing data handled? Were they presumed to be absent?

For the prospective data collection, the information collected on the case record form were compared against the patient electronic medical record and were found to be highly concordant - sometimes the negative information in the patient record was recorded as eg 'no dysuria' and sometimes the feature was not mentioned if it was negative. For the retrospective data collection we therefore concluded that missing data indicated an absence of the feature.

This has been added to the Methods as follows: 'The concordance enabled missing clinical features in the retrospective clinical record review to be presumed absent.'

6. What was the standard route of urine collection by age?

The standard route of urine collection was clean catch in younger children and mid-stream urine in older patients. Occasionally young patients had urine collected by in/out catheter or supra pubic aspiration, but bags were never used.

This has been added to the box 1 as follows: '...collected by clean catch, mid-stream urine, in/out catheter or supra-pubic aspirate.'

7. I question the need for a separate analysis of patients with "probable UTI" using a definition that did not include positive culture. Although the colony count threshold may differ by the collection method and likelihood of contamination, if there is no bacterial growth, how is it a probable UTI? Given that the signs and symptoms of UTI overlap with other disease entities and that the specificity of pyuria alone is suboptimal, understanding the score performance in this population may not be useful.

We thought carefully about this too, and it was a deliberate methodological choice. Patients were indeed excluded from the study if they had pyuria and clearly had another infection. However, there was a group of patients with NICE-guided symptoms of UTI and pyuria who subsequently had no growth due to antibiotics being given prior to urine collection. We deliberately separated this group for subsequent analysis so that they did not spuriously affect the development or validation of the clinical score. It allowed us to test the score on this group of children diagnosed and managed as having UTI

in ED when decisions need to be made prior to receiving urine culture results. This is the real world situation and provides an empiric practical solution.

Results

1. Of the 9 children de-escalated from IV to PO, how many doses of IV did they receive in the ED? It would be helpful to understand the clinical characteristics of this group to determine if the designation of “oral antibiotics applicable” is appropriate. If this cannot be ascertained, I would add this as a limitation.

Thank you for this suggestion. We have added to the manuscript as above to describe the antibiotics received by this group of 9 patients. Additionally, we further analysed the data and have briefly described the clinical characteristics of these 9 patients in the manuscript as follows: ‘For the 9 who de-escalated from IV to oral by 24 hours, 2 had a score of 1 (both recurrent UTI), 4 scored 2 (2 fever and vomiting, 1 fever and tachycardia, 1 fever and urological abnormality), 1 scored 3 (fever, rigors and vomiting) and 2 scored 4 (both fever, vomiting, recurrent UTI and urological abnormality – 1 with neurogenic bladder, 1 with pyeloplasty).’

2. How many children with recurrent UTIs had urologic abnormalities?

We have further analysed these data and added to the manuscript as above.

Tables and Figures

Table 1. Please define urologic abnormality for the reader in the footnote.

This has been done and also in box 1 as above.

Flow diagram-what proportion of children were collected prospectively vs. retrospectively?

We have added the proportions in each group to the Figure 1a as suggested.

Discussion

1. Page 12, line 26-I may also add that this study excluded infants from the derivation population, which may be another explanation for the poor performance in addition to the noted variability in care. While we agree that this is possible, the score was also tested on other populations not in the derivation cohort and was successful so we don’t believe this is the major factor compared to the variability in care in the under 1s. We would be happy to include it as a possibility alongside a discussion about the discrepancy between the score’s success in other test populations not in the derivation cohort compared to the under 1s, if the editors were able to give us some license on the total word count.

Reviewer 2

General comment

The authors of this manuscript reported the use of a derived, validated and tested clinical score (RUPERT score) to guide the initial route of antibiotics in children with urinary tract infection (UTI) who presented at an Emergency Department. Given the long-term renal complications associated with poorly treated UTI, it is imperative to initiate the appropriate antibiotic therapy early, using the right antibiotic choice and the right route of administration. This fact underscores the importance of this study.

Thank you.

However, there are specific concerns I have about this manuscript which need clarifications/ corrections from the authors.

Specific comments

1. Abstract: Under ‘Objectives’, the introductory statement is unclear. I think the authors need to clarify which form of UTI that can be managed with oral antibiotics. I suggest they specify that ‘most children with uncomplicated UTI can be managed with oral antibiotics.’ Again, the next statement is befuddling as it failed to differentiate ‘complicated UTI’ from ‘UTI with complications’. Both terms are not the same. The title of the manuscript rather suggests that the authors may mean the former (‘complicated UTI’). Thus, the statement that ‘identifying those likely to fail oral and need intravenous (IV) antibiotics due to complications is challenging’ may better read ‘identifying those likely to fail oral or need intravenous (IV) antibiotics due to complicated UTI is challenging.’

Although we wanted to start with the true statement that most of all UTI can be managed with oral antibiotics, we agree with the reviewer highlighting that this creates unnecessary lack of clarity. As suggested we have changed the first sentence as follows: 'Most children with uncomplicated urinary tract infections (UTI) can be managed with oral antibiotics.'

For the second point, we agree that this could be clearer. We have not used the term 'complicated UTI' in this sentence because there is no unifying definition of complicated UTI (we wish there were as this study would have been much easier!) and definitions variably include multiple different complicating factors related to past history and current presentation. However, 'UTI with complications' could be erroneously construed as meaning 'UTI that subsequently has complications during progress'. We have therefore changed the second sentence as follows: 'However, identifying those likely to fail oral and need intravenous (IV) antibiotics due to complicating features at presentation is challenging.' We have made the same change in the first paragraph of the introduction.

Under 'Design', the study design was not clearly stated. If this was a cohort study, what was the intervention and what outcomes were measured?

Thank you for pointing this out. Although we used a large cohort of children, this is not a traditional cohort study and there is not a traditional intervention to compare to a population that did not receive the intervention. We have changed the word 'cohort' to 'population'.

Under 'Results', the authors reported that 1,240 patients were included. It is not correct to commence the statement with Arabic numerical; the number should have been written in words.

We have added 'A total of' before the number.

More importantly, the authors should clarify what is meant by the finding that a RUPERT score of ≥ 3 guided the commencement of IV antibiotics that accurately classified 80% of the patients. Do they mean accurately classifying them as having complicated UTI? The same clarification is needed for the succeeding statement on 168 patients in the validation cohort.

We agree this needs clarification. Using the cut-off of ≥ 3 for IV antibiotics (and therefore ≤ 2 for oral antibiotics) accurately classifies the route of antibiotics (both IV and oral) after 24 hours for 80% of patients. We have clarified this in the Abstract and Results as follows: '...correctly classified the route of antibiotics after 24 hours for 80% of patients'

Complicated UTI has previously been defined as UTI likely to fail oral treatment, so this score is also a potential way to define complicated UTI as explored in the Discussion. However, the reviewer's point is helpful for the reader, so to introduce this concept earlier than the Discussion, we have added to the text in the Results as follows: 'This threshold for use of IV may act as a surrogate for identifying more complicated UTI.'

Finally, ensure that your keywords are MeSH-optimized.

We have done this: urinary tract infection, pyelonephritis, paediatric A&E and ambulatory care, paediatric infectious disease and immunisation.

1. Introduction: From the fourth to sixth sentences, the authors should clarify whether they meant complicated UTI or complications of UTI. While the former is associated with underlying conditions such as urological anomalies, bladder outlet obstruction, immunosuppression etc., the latter comprises sepsis, recurrence of infection, Urosepsis, renal scarring etc.

We agree that clarification is needed. Because of the varying definitions of complicated UTI (including the definition above), we have steered away from the term to avoid confusion. The premise of the study is that it is not binary between uncomplicated and complicated. With this clinical score, we have used the fact it is not binary to develop a scale with a threshold to help use in practice. With the greatest respect to the reviewer, we therefore think it is important to leave the terminology as is. We have provided further exposition at the end of this response about the lack of clarity around the definition of complicated UTI based on a recent systematic review for the European Society of Paediatric Infectious Diseases Complicated UTI Guideline involving the senior author (PAB).

We have added a sentence to the introduction to make this much clearer as follows: 'There is no consensus on the definition of complicated UTI: definitions variably include multiple factors related to past history and current presentation that may result in divergence from typical management.' Except where 'complicated UTI' has been used by others, we have changed the text from 'complications of UTI' in the Introduction and elsewhere to 'complicating features at presentation'.

1. Methods: The authors stated here that the study design was a clinical cohort of consecutive patients presenting at the Emergency Department. It is presumed that a cohort study is prospective in nature. However, the authors reported that 'to include sufficient patients to validate the findings, participants were also identified retrospectively through diagnostic coding of UTI.' Can they please clarify this?

Agreed – to correct this we have amended the Methods to use the word 'population' instead as above. We have also clarified the wording around the prospective and retrospective nature of the data collection as follows: 'Patients were enrolled both prospectively to ensure integrity of the information collected, and also retrospectively to include sufficient patients to validate the findings.'

Again, why was immunodeficiency listed as one of the exclusion criteria of the study given that it is one of the features that also defines complicated UTI?

This is a good question - some guidelines include immunocompromise in the complicated UTI definition while others do not. After considering this carefully, we determined that patients with immunodeficiency presenting to ED are usually treated under other guidelines (e.g. febrile neutropenia guidelines) that supersede the guidance for managing them with UTI. If they are not currently neutropenic there is no absolute need for IV antibiotics, so it was important that this didn't erroneously skew the results. We have clarified the reason in the manuscript as follows: 'Patients were excluded if they were treated in ED via superseding pathways: severe infection (e.g. sepsis or meningitis) or immunodeficiency (e.g. febrile neutropenia or post renal transplant).'

Under 'clinical procedure and patient follow-up', what do the authors mean by collecting data from retrospective patients since the study was essentially a prospective one? (Line 5).

We have now clarified as above regarding the combined prospective and retrospective nature of the study.

1. Results: The authors reported that 'Of the 1,240 with UTI, 276 (22%) were aged 3m-11m, 831 (67%) were 12m-11y and 133 (11%) were 12y-17y'. I assume these were prospectively recruited participants. What then is the number of the retrospective participants mentioned previously?

Among the latter, was the diagnosis of UTI similarly made with the NICE diagnostic criteria?

Yes both prospectively and retrospectively enrolled patients all fulfilled NICE criteria, which is now clear in the Inclusion/Exclusion criteria as above.

We have also added the proportion of prospective and retrospective patients to the flow diagram of patient numbers as above.

Figures 1 and 3 need magnification for clarity and understanding of the findings.

We wonder if this is a technical issue as the uploaded figures conform to guidelines regarding size and dpi.

1. Discussion and conclusion: Well presented with no ambiguity.

Thank you.

1. References: Although much of the cited publications appear relevant, some are very old publications. For instance, reference number 1 may well be replaced by a more recent publication. There is no consistency in writing the citations; while some journal names were written in their index medicus titles, others appeared with their full names.

This is a fair comment – we wanted to be acknowledge the work of earlier researchers on whose work this study was built. We have added some relevant more recent studies. The inconsistency in references has been corrected.

Reviewer 3

The authors have tried to develop a score to help with the decision if to treat children with a UTI with iv or oral antibiotics. They include children with all kinds of UTIs and of virtually all ages. I am sorry but I do not believe that this is a workable concept. The range of different conditions within that spectrum is too wide to make it meaningful.

We value this comment because the range of conditions which broadly and inconsistently fall under the wide umbrella of 'complicated UTI' is exactly the problem we are addressing in this study. Complicated UTI has consistently been put in the 'too hard basket' and excluded from trials, guidelines and even retrospective studies, as shown in the manuscript. Clinicians are left trying to determine if their patient really has a complicated UTI, what exactly that means and how to manage the child in front of them. Lacking clear guidance, they may default to IV use if they are not sure and the patient is febrile, despite a Cochrane review (Strohmeier, 2014) concluding decisively that fever alone is not a reason for using IV antibiotics. The study both avoids the controversies of 'what defines a complicated UTI' (see above) and has explored the concept of standardising management by deliberately using multiple different components of the wide spectrum of UTI. The entry point of the study is not complicated UTI – it is all UTI, so that clinicians can build on the clinical features that the child presents with and determine whether they reach a threshold for considering IV antibiotics. As explored in the Discussion (and in response to Reviewer 2's comments, now introduced in the Results), this may enable us to think more laterally about combining different complicating features to assess a presentation as 'more complicated' rather than the binary 'uncomplicated' versus 'complicated'. As shown in this study (and reflecting the reviewer's comment here), many children with so-called complicated UTI are treated with oral antibiotics, so assigning this term does not help guide whether a child should be managed with IV or oral antibiotics.

It is difficult to see but an exceptional child with an afebrile UTI that would need anything else but oral antibiotics. This group should thus be excluded from the study.

The inclusion criteria were all children being treated primarily for UTI rather than children with complicated UTI. Since fever alone is not a reason for IV use as the reviewer highlights, we were interested in the group who were afebrile. As a tertiary centre, we potentially see more than the average number of exceptional children, but this strengthens the findings as most studies have too few children with complications to draw any conclusions. In the derivation cohort, there were only 3 afebrile children who remained on IV antibiotics after 24 hours: 2 with a RUPERT score of 3 (both with moderate to severe urological abnormalities – neurogenic bladder, mitrofanoff, severe VUR – and recurrent UTI, 1 also with vomiting and 1 also with rigors) and 1 with a RUPERT score of 1 (with recurrent UTI, on prophylaxis and additionally had had 7 days of oral treatment with a different antibiotic already – in reality this child needed an antibiotic with a different susceptibility spectrum rather than IV antibiotics, but these culture results were not available until 48 hours). We have further analysed the whole population of 1240 children, and found that of 514 children who were afebrile, a minority of 72 (14%) received initial IV antibiotics, half of whom switched to oral within 24 hours – 94% of those had a RUPERT score of 2 or under: if applied prospectively these patients would all have been recommended for initial oral antibiotics. Of those who continued IV beyond 24 hours, only 24% had a RUPERT score of 3 or more, and the remainder would have been recommended for oral. The score is effective in this group as without fever, they have to have 3 other components for IV to be recommended, and this decreases the likelihood that they will unnecessarily be prescribed IV. This concurs with the reviewer's statement that a minority of afebrile children should receive IV and this score allows that to be standardised. We could add an analysis of afebrile patients if the reviewers would prefer.

Any child with signs of a potential sepsis, like rigor and tachycardia out of proportion to the temperature, should also be evaluated on an individual basis and most likely in nearly all cases be given iv antibiotics. (And intensive care?)

We completely agree that children severely unwell with sepsis should receive IV antibiotics, so these children were excluded from the study because they follow a different management pathway in ED - our institutional 'Sepsis pathway' which recommends initial IV antibiotics for all. This is included in the Methods and has been clarified as above in response to Reviewer 2. None of the patients included in the study required intensive care.

Like this reviewer, we were curious to determine how children outside of this severe sepsis group, but with features of potential sepsis, were managed. We have further analysed the data and of the whole population, 49/99 (49%) patients with rigors or a history of rigors were treated with oral antibiotics, 719/903 (80%) patients who were tachycardiac in ED had oral antibiotics (including both tachycardia while febrile and while afebrile), and 25/45 (56%) patients who had both features were treated with oral antibiotics without switching to IV after 24 hours. From our own clinical experience we suspected this, and this study confirmed in a data-supported way that there is no universal clinical feature associated with IV use. Our experience of the use of oral antibiotics despite individual complicating features (both on patient history and clinical presentation) was one of the starting points for this study. It led us to test the hypothesis that multiple features combined may enable a more accurate and standardised method of proposing the initial route of antibiotics.

We also agree that patient care should always be individualised, and have made clear in the conclusion that this score should be used to guide initial route of treatment, not be treated as a mandate.

We have added to the Methods as above: 'Patients were excluded if they were treated in ED via superseding pathways: severe infection (e.g. sepsis or meningitis) or immunodeficiency (e.g. febrile neutropenia or post renal transplant).' We have added to the Discussion: 'Scores in the lower range reassure that even with features potentially associated with sepsis (rigors, tachycardia), in isolation do not necessitate IV antibiotic use.'

Children with urinary tract malformations are a heterogenous group with mild malformations like mild hydronephrosis to severe posterior urethral valves with markedly impaired kidney and bladder function. These children need an individualised approach taking a number of important variables into account.

We agree that there is a wide spectrum of urinary tract malformations. The categorisation we used reflects the previous RCTs of IV versus oral treatment which simply state 'urological abnormality' as an exclusion. As above, the entry point for this study was 'all UTI' not 'complicated UTI' and the aim was to keep this as open and as simple as possible, and that the categorisation is documented by the treating physician as deemed relevant for the management of the individual child's UTI. We have clarified by defining urological abnormality more clearly in the text, box 1 and table 1 as above.

We also agree with the reviewer that an individualised approach should be used, and at the moment there is a lack of evidence to provide a basis for such an individualised approach. This score aims to decrease unwarranted variation in care by using a child's individual features to build to a guide for that individualised approach.

We have added to the Discussion as follows: 'It allows for an individualised approach by using an individual child's own clinical features rather than trying to impose a particular definition of complicated UTI, which is additionally lacking evidence for management.'

VERSION 2 – REVIEW

REVIEWER	Tamar Lubell New York-Presbyterian Morgan Stanley Children's Hospital, Department of Emergency Medicine
REVIEW RETURNED	22-Mar-2024
GENERAL COMMENTS	The authors responded thoughtfully to the reviewer's questions and incorporated suggestions when possible. I appreciate the shift in the wording to "UTI with complicating features at presentation," as I

	think it better encompasses the population the authors are trying to assess. There remain concerns regarding the generalizability of the study findings owing to the heterogeneous population (from a wide ranges of ages and underlying GU abnormalities). Additionally, though the information provided about the cross-over group from IV to PO is appreciated, I have concerns with the presumption that an initial dose of IV antibiotics (q24 or q6h dosing, which does not include other potential forms of therapy via IV), with a subsequent switch to oral at 24h indicates that oral antibiotics on presentation were appropriate, particularly given that the majority of these children had complicating features on presentation. The study cited as support for this assumption was for cellulitis. It is unclear if this applies to other bacterial infections like UTIs. Despite these concerns regarding the generalizability and the potential impact of the study findings, the authors were generally careful not to overstate the significance. This study may serve as a starting point to better define complicated UTI (or UTI with complicating features) and may help the clinicians understand how a combination of features may serve to guide the initial route of antibiotics. Further prospective validation in other populations and further score refinement would be of utility to clinicians. Additional points needing revision: Tables and Figures: The uploaded Supplemental Tables 1 and 2 appear to be the same document. If this was an error, the correct file should be uploaded for review.
--	--

REVIEWER	Samuel Uwaezuoke University of Nigeria Teaching Hospital
REVIEW RETURNED	07-Mar-2024

GENERAL COMMENTS	Thank you for adequately addressing my concerns and questions
---

VERSION 2 – AUTHOR RESPONSE

Reviewer 1

The authors responded thoughtfully to the reviewer's questions and incorporated suggestions when possible. I appreciate the shift in the wording to "UTI with complicating features at presentation," as I think it better encompasses the population the authors are trying to assess.

Thank you.

There remain concerns regarding the generalizability of the study findings owing to the heterogeneous population (from a wide ranges of ages and underlying GU abnormalities).

We considered that the findings would be more generalizable specifically due to the heterogeneity of the population, including the fact we included uncomplicated UTI for specificity, rather than if we had targeted a more homogenous group. If populations are heterogenous in different ways (eg in a rural hospital rather than our tertiary hospital) then this could affect generalizability, hence the need for prospective multicentre validation. We have added the following to the limitations, but if we have misunderstood the reviewer's intent then please let us know.

'Although the heterogeneity of our population provides some assurance about generalizability, the score could perform differently in other populations that are more homogenous or heterogenous in a different way. Prospective validation in other settings is therefore essential.

Additionally, though the information provided about the cross-over group from IV to PO is appreciated, I have concerns with the presumption that an initial dose of IV antibiotics (q24 or q6h dosing, which does not include other potential forms of therapy via IV), with a subsequent switch to oral at 24h indicates that oral antibiotics on presentation were appropriate, particularly given that the majority of these children had complicating features on presentation. The study cited as support for this assumption was for cellulitis. It is unclear if this applies to other bacterial infections like UTIs.

Regarding using the presumption that IV antibiotics for <24 hours to mean that IV antibiotics were likely not needed, we have previously shown that the patients who had <24 hours IV antibiotics for UTI were more clinically similar to patients who only received oral antibiotics than patients who received 2 or more days IV antibiotics. We have add this reference, which at this stage is a conference abstract. We have added the following to the Discussion:

‘Although the methodology was devised for cellulitis, we have previously shown that patients with UTI who received less than 24 hours’ IV antibiotics were more similar to patients who received only oral antibiotics than patients who received 2-3 days’ IV antibiotics[ref]. This adds weight to the supposition for the devised gold standard that this group only needed oral antibiotics, although without a cast-iron clinical marker for ‘need for IV’, this could not be guaranteed in every case.’

Despite these concerns regarding the generalizability and the potential impact of the study findings, the authors were generally careful not to overstate the significance. This study may serve as a starting point to better define complicated UTI (or UTI with complicating features) and may help the clinicians understand how a combination of features may serve to guide the initial route of antibiotics. Further prospective validation in other populations and further score refinement would be of utility to clinicians.

We like the concept of a starting point and score refinement so have added the following to the Discussion: ‘However, it aims to be a starting point for standardising the rationale for IV antibiotics’ and ‘The score will now benefit from external validation, refinement and impact analysis.’

Additional points needing revision:

Tables and Figures:

The uploaded Supplemental Tables 1 and 2 appear to be the same document. If this was an error, the correct file should be uploaded for review.

Apologies – this was an error as the supplemental/online tables are included in the main manuscript for review.